# EXPLORATION FOR DEPLOYMENT-EFFICIENT REINFORCEMENT LEARNING AGENTS

## ABSTRACT

Reinforcement learning (RL) provides a rich toolbox with which to learn sequential decision making policies. Notably, the ability to learn solely from offline interaction data has been a highly successful modality for training real-world policies. However, a gap exists in this paradigm when the offline dataset does not cover all the behaviors necessary to extract optimal policies. Naively, one can pre-train a policy using offline RL and fine-tune it using online RL; this can lead to catastrophe in settings like healthcare and autonomous driving, where deploying an unverified policy is irresponsible. Deployment efficient learning is a potential solution, where the number of distinct data collection policies is relatively low compared to the number of updates to the policy. We argue that safely improving a dataset requires a deployment efficient algorithm with a carefully constructed data collection policy. We introduce a framework with a stationary exploration policy that aims to reduce out-of-distribution uncertainty while maintaining strong returns. We establish theoretical guarantees of this exploration framework without finetuning and demonstrate our method on a large-scale supply chain environment with real-world data.

## 1 INTRODUCTION

Offline RL algorithms are popular in domains where online interactions can be costly or unsafe (Achiam & Amodei, 2019; Hu et al., 2024; Liu et al., 2023). To avoid potentially dangerous states, many offline algorithms (Kumar et al., 2020; Kostrikov et al., 2021) regularize the learned policy to take actions that stay within the distribution of interactions within the dataset. Naturally, the performance of policies trained with offline RL algorithms depends significantly on the coverage of this offline data.

Offline-to-online RL algorithms seek to overcome this limitation by refining offline trained policies via online interaction with the environment (Nair et al., 2020; Mark et al., 2024; Ball et al., 2023; Wang et al., 2023; Zhou et al., 2025). However, this approach falls short in the real world due to practical considerations surrounding the cost and safety of online exploration. Real-world applications like supply chain and autonomous driving (Gottesman et al., 2019; Kiran et al., 2021) have little tolerance for error and typically require policies to be verified and tested before deployment. For this reason, updating the policy parameters online, as most offline-to-online methods do, cannot be permitted in the real world as every update to the policy parameters could lead to unverified behavior. As testing and verification can be time consuming and expensive, it makes sense to use the number of policy deployments as a suitable measure of efficiency. Specifically, many real world systems require a more *deployment efficienct* (Matsushima et al., 2020) algorithm, i.e. a low ratio of the number of unique policies deployed in the environment during training to the number of samples collected. On-policy RL methods are at one extreme of the deployment efficiency spectrum, requiring one deployment per update. Purely offline methods are at the other end, only utilizing one deployment in total (by whatever mechanism collected the training dataset). In real-world settings, the only feasible way to improve an offline-trained policy is to explore in a manner that is neither costly in terms of deployment efficiency nor leads to losses due to poor selection of actions.

A high-profile and socio-economically relevant setting is that of supply chain management (Madeka et al., 2022), where an agent must place inventory orders to vendors to stock a company's warehouses. In this domain, state-of-the-art RL agents (Andaz et al., 2023) are trained offline using historical

data collected from a traditional linear programming-based behavior policy. However, to improve the offline policy, it is necessary to collect quality online interactions through exploration in the real world, but each suboptimal action has severe financial overheads. Since policies need to be verified and back-tested for safety and performance (Corsi et al., 2024; Matsushima et al., 2020; Madeka et al., 2022; Amir et al., 2021) before deployment, it is difficult to justify continuous online finetuning. Instead, we propose to use a stationary exploratory policy to collect high quality samples that can be used to re-train the policies offline on the *augmented* dataset. This stationary exploration policy should satisfy two desiderata: (a) it should *not be costly*, meaning that executing the policy in the real world should not lead to significant costs, i.e. the policy should not deviate a lot from the dataset policy, and (b) its exploration should be *targeted*, meaning that it should carefully collect data that are close to the known safe regions, while still meaningfully augmenting the dataset for high-quality offline policy improvements for future deployments.

We propose a framework to address the largely overlooked challenge of real-world exploration. Specifically, we design an exploratory policy to visit states and actions beyond the existing dataset, thereby reducing suboptimality relative to the current policy. We express the suboptimality of an offline dataset as a function of the set of states visited by the optimal policy but not contained in the dataset. We construct an exploratory policy that directly minimizes this suboptimality by exploring to include these states in the offline dataset. Intuitively, it is only these states can affect the performance of the policy trained from the dataset and should be the only states that we should care about while exploring. Our exploratory policy balances two competing objectives: learning a policy that stays within the support of the static dataset, while collecting novel data that would improve the usefulness of the offline dataset for future deployments. We compare our exploration strategy against commonly used exploration strategies like $\epsilon$-greedy and count-based exploration on a discrete navigation environment. We empirically evaluate our exploration strategy on a real-world supply chain system (Madeka et al., 2022; Andaz et al., 2023).

Our contributions can be summarized as: (1) establishing a framework for exploring through online interactions without fine-tuning the policy online, (2) developing a principled exploration strategy from the offline dataset that can collect informative samples through online interactions, and (3) empirically analyzing the samples collected using our exploration strategy on a discrete navigation environment and a real-world supply chain application (Madeka et al., 2022; Andaz et al., 2023).

## 2 RELATED WORK

**Offline to Online finetuning:** Offline RL algorithms are essentially off-policy RL algorithms that add additional constraints to the algorithm to ensure that the policy always stays within the data support. A number of such algorithms explicitly constrain the value functions (Kumar et al., 2020; Yu et al., 2020; Kidambi et al., 2021) or use regularization to ensure that the policy stays within the data support (Wu et al., 2019; Nachum et al., 2019). In all cases, the performance of the policy is restricted by the dataset quality (Kidambi et al., 2021).

Online finetuning has been extensively used (Nair et al., 2020; Mark et al., 2024; Ball et al., 2023; Wang et al., 2023) to mitigate the dataset bias. While the offline RL algorithms themselves can be directly used for online finetuning like in Peng et al. (2019a); Nair et al. (2020); Kumar et al. (2020); Kostrikov et al. (2021), they are overly pessimistic. A number of works have been introduced that simply reduce the regularization for conservatism during online finetuning (Lee et al., 2021; Zhang et al., 2023; Hong et al., 2023; Ball et al., 2023; Zheng et al., 2023). There have also been works that have used offline datasets as a prior for online RL algorithms (Mark et al., 2024; Ball et al., 2023).

All these methods have one major assumption: there is no restriction on online finetuning. The focus of all these works is to produce good policy updates combining the online samples and offline dataset. As discussed in works like Matsushima et al. (2020), there is often costs associated before deploying any policy in the real world. Our work differs from all these offline-to-online methods in that it aims to develop an exploration scheme suitable for applications where it is not possible to deploy arbitrary policies.

**Exploration:** Our method constructs an exploratory policy from batched experience to deploy in the real world. Here, we will discuss some commonly used methods for constructing exploratory policies. Epsilon greedy (Watkins, 1989; Auer et al., 2002) is a simple, yet highly effective method used to take exploratory actions with probability $\epsilon$. The idea is to ensure that the policy has some entropy and chooses every action with a non-zero probability. Another way of increasing the entropy of the policy

is through explicit entropy maximization as in the MaxEnt RL methods (Haarnoja et al., 2017; 2018).

Selecting actions using an upper confidence estimator for the average return (Q-function in case of RL algorithms) have been inspired from the UCB algorithm in multi-arm bandits (Auer et al., 2002). Strehl & Littman (2008) proposed using an uncertainty bonus for the reward function providing convergence guarantees. Algorithms have been proposed that extend the uncertainty bonus in several ways: using pseudocounts (Bellemare et al., 2016; Lobel et al., 2023), model errors (Burda et al., 2019) and using ensembles to estimate uncertainty (Kidambi et al., 2021). Distribution matching approaches have also been proposed that aim to increase the entropy of the state-visitation distribution (Lee et al., 2019; Agarwal et al., 2024). While our exploration method also draws from a similar objective, these exploration strategies have been proposed in settings where the aim is to explore all states, ultimately leading to a nearly zero uncertainty bonus for all states. Our method, on the other hand, uses the uncertainty bonus to select a few states from the prior policy distribution.

## 3 PRELIMINARIES

We consider problems that can be modelled as Markov Decision Processes (MDPs) (Puterman, 1990), which are defined as $\mathcal{M} = \langle \mathcal{S}, \mathcal{A}, P, r, \gamma, \rho_0 \rangle$ where $\mathcal{S}$ is the state space, $\mathcal{A}$ is the action space, $P : \mathcal{S} \times \mathcal{A} \longmapsto \mathcal{S}$ is the transition probability function, $\gamma \in [0, 1)$ is the discount factor, $\rho_0$ is the initial state distribution and $r : \mathcal{S} \times \mathcal{A} \longmapsto \mathbb{R}$ is the reward function. A policy $\pi_\theta : \mathcal{S} \longmapsto \mathcal{A}$ represents the probability of taking an action $a$ from state $s$. The policy induces a trajectory distribution $p^\pi(\tau) = \rho_0 \Pi_{t=1}^\infty \pi(a_t|s_t)P(s_{t+1}|s_t, a_t)$ and a corresponding state-visitation distribution $d^\pi(s) = (1 - \gamma)\sum_{t=0}^\infty \gamma^t p^\pi(s_t = s)$. The expected return is defined as $J(\pi_\theta, \mathcal{M}) = \mathbb{E}_{\tau \sim p^\pi(\tau)}[\sum_{t=0}^\infty \gamma^t r(s_t, a_t)]$. The goal of the agent is to find the optimal policy $\pi^*$ that maximizes the expected return, i.e,. $\pi^* = \arg\max_\pi J(\pi, \mathcal{M})$.

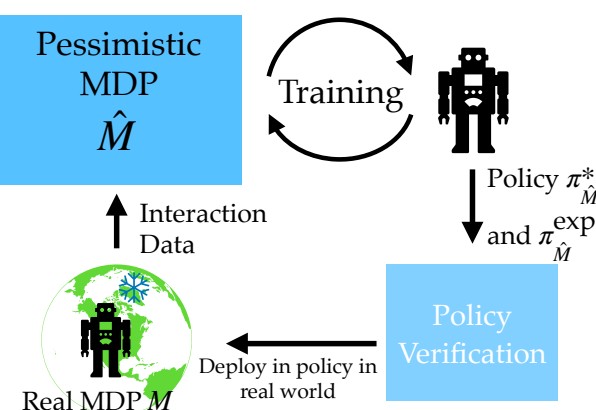

Figure 1: **Real-world RL deployment life cycle:** The real world represents an MDP $M$ which is unknown. A pessimistic MDP is constructed using interactions from $M$. Policies are trained on $\hat{M}$ which are verified and tested before deploying in the real world.

Offline RL algorithms try to regularize the policy to avoid taking actions out of dataset support. The regularization can be in the form of pessimism in Q-value for out-of-distribution actions (Kumar et al., 2020; Shimizu et al., 2024) or pessimism in the reward function based on uncertainties in the model predictions (Yu et al., 2020; Kidambi et al., 2021). In some sense, it can be assumed that the general objective for offline RL algorithms are,

$$\max_\pi \quad J(\pi, \mathcal{M})$$
$$s.t. \quad D(p^\pi(s, a)||\hat{p}(s, a)) \leq \epsilon, \tag{1}$$

where $D$ is written as a divergence, but can more generally be any measure of the difference between distributions, $p^\pi(s, a)$ is a measure of the true likelihood of state-actions under the policy, and $\hat{p}(s, a)$ is a measure of the likelihood of the state-action in the dataset (Nachum et al., 2019).

## 4 THE REAL-WORLD EXPLORATION FRAMEWORK

We formally introduce the real-world deployment cycle in Figure [] and illustrate where exploration fits in here. Given a Real-World MDP $\mathcal{M}$, data is collected to create an offline dataset which implies an MDP $\hat{\mathcal{M}}$. Successful offline RL algorithms (Kostrikov et al., 2021; Kidambi et al., 2021) have introduced pessimism in the MDP $\hat{\mathcal{M}}$ to ensure that the policy does not steer away from the data

distribution. An RL policy is trained on this pessimistic MDP $\hat{\mathcal{M}}$ to produce a policy $\pi$. This $\pi$ is verified and tested through a series of application specific protocols and deployed for execution in the real world.

To explore in the MDP $\mathcal{M}$, an exploratory policy $\pi_{exp}$ is constructed and needs to go through the similar verification and testing protocols before it can be executed in the real world. This implies that $\pi_{exp}$ cannot be arbitrarily updated during execution. Every action taken in the real-world incurs costs and safety considerations. Since $\pi_{exp}$ takes over the execution from policy $\pi$ $\pi_{exp}$ cannot be expected to run for long in the environment. Additionally, this also implies that $\pi_{exp}$ cannot take highly suboptimal actions or deviate significantly from $\pi$. For instance, in an autonomous driving scenario, when $\pi_{exp}$ cannot execute random actions that can lead to catastrophes on the street. Similarly, in a trading application, $\pi_{exp}$ cannot make random trades that can incur massive costs. Overall, the criteria for constructing $\pi_{exp}$ are (1) $\pi_{exp}$ should have a cost/return close to $\pi$, (2) since $\pi_{exp}$ wont be executing for long, it needs to target meaningful regions of the state space and (3) it cannot be updated online. These desiderata are often ignored while looking at exploration in simulation or simulated real-world settings.

## 5 METHOD

The success of exploring in the real world hinges on two key considerations. First, the policy must selectively collect data that will augment the offline dataset in such a way that it improves future policy deployments. Second, the exploration conducted online must be controlled, meaning that it should stay close to the support of the offline data. In this section, we present the theoretical formulation of our efficient exploration framework, along with practical approximations.

### 5.1 THE EXPLORATORY POLICY

First, we construct the exploratory policy given a dataset $\mathcal{D} = \{(s_i, a_i, s_i', r_i)\}_{i=1}^N$; this is done using a distribution matching objective against a target distribution. First, we formalize the sub-optimality of a policy obtained using an offline dataset. We define $u(s, a) \approx \log p((s, a) \notin D)$ to provide a measure of the likelihood of $(s, a)$ not being in the dataset. Using $u(s, a)$ we can formally define the uncertain set (Kidambi et al., 2021):

**Definition 5.1.** Let $\mathcal{U}_\mathcal{D}$ be the set of state-action pairs $(s, a)$ that are not contained in the dataset $D$. Let $u(s, a)$ represent the likelihood of $(s, a) \notin \mathcal{D}$. Define $\mathcal{U}_\mathcal{D} = \{(s, a)|u(s, a) > \alpha\} \approx \{(s, a) \in \mathcal{S} \times \mathcal{A}|(s, a, ., .) \notin \mathcal{D}\}$, where $\alpha$ is a scalar parameter.

If $(s, a) \in \mathcal{U}_\mathcal{D}$, $p^\pi(s, a)$ would be low and if the probability $\pi(a|s)$ is high, then $D(p^\pi(s, a)||\hat{p}(s, a))$ would be high. In other words, creating this partition in the state space allows the creation of an approximation of the given MDP $\mathcal{M}$ where the expected return is a pessimistic estimate of the objective in Equation (1). This pessimistic MDP will ensure that the trained policy will restrict the policy from taking actions outside the support of the dataset (Kidambi et al., 2021).

**Definition 5.2.** Given an MDP $\mathcal{M} = \langle \mathcal{S}, \mathcal{A}, P, r, \gamma, \rho_0 \rangle$, a dataset $\mathcal{D} = \{(s_i, a_i, s_i', r_i)\}_{i=1}^N$, define a pessimistic MDP $\mathcal{M}_\mathcal{D} = \langle \mathcal{S} \cup HALT, \mathcal{A}, P_p, r_p, \gamma, \rho_0 \rangle$ where,

$$P_p(s'|s, a) = \begin{cases} \delta(s' = HALT), & (s, a) \in \mathcal{U}_\mathcal{D} \\ P(s'|s, a), & \text{otherwise} \end{cases} \qquad r_p(s, a) = \begin{cases} -R_{max}, & (s, a) \in \mathcal{U}_\mathcal{D} \\ r(s, a), & \text{otherwise} \end{cases}$$

Additionally, we make the following assumption to derive the suboptimality gap in the offline data. This assumption is also made by most offline RL algorithms using Equation (1) as their objective. It simply states that the offline RL algorithm produces the optimal policy for the pessimistic MDP $\mathcal{M}_\mathcal{D}$.

**Assumption 5.3.** Let $\pi_D^*$ be the best policy obtained from a dataset $\mathcal{D}$, essentially by optimizing the objective in Equation (1). We assume that $\pi_D^*$ maximizes the expected return in $\mathcal{M}_\mathcal{D}$ i.e. $J(\pi_D^*, \mathcal{M}_\mathcal{D}) \geq J(\pi, \mathcal{M}_\mathcal{D}) \quad \forall \quad \pi$.

Let $d^\pi(\mathcal{X})$ be the visitation of set $\mathcal{X}$ following policy $\pi$ and using the true dynamics $P$. We can now bound the true returns of the best policy obtained using the offline RL objective eq. (1) with the return of the optimal policy on MDP $\mathcal{M}$ or in other words define the suboptimality gap in the offline dataset.

**Theorem 5.4.** *Let $\mathcal{M}$ be a given MDP and $\mathcal{D}$ be the offline dataset. Let $\pi^* = \arg\max_\pi J(\pi, \mathcal{M})$ be the true optimal policy on $\mathcal{M}$ and $\pi_D^* = \arg\max_\pi$ Equation (1) $\approx \arg\max_\pi J(\pi, \mathcal{M}_\mathcal{D})$ be the best offline RL policy on dataset $\mathcal{D}$. The following holds: $J(\pi^*, \mathcal{M}) - J(\pi_D^*, \mathcal{M}) \leq \frac{2R_{max}}{(1-\gamma)^2} d^{\pi^*}(\mathcal{U}_\mathcal{D})$.*

A direct consequence of Theorem 5.4 is that if $d^{\pi^*}(\mathcal{U}_\mathcal{D}) = 0$, i.e. all transitions taken by the optimal policy are in the dataset $\mathcal{D}$, the optimal return of the best offline policy and the true optimal policy are the same. In other words, the best offline policy is in fact the true optimal policy. In general, if we want to improve the dataset, we need to tighten the bound or minimize $d^{\pi^*}(\mathcal{U}_\mathcal{D})$. This means the exploratory policy must aim to maximize this visitation.

Formally, let $\mathcal{U}_\mathcal{D}^*$ be the set of state-action pairs visited by the optimal policy $\pi^*$ which are in the set $\mathcal{U}_\mathcal{D}$. Mathematically, $p((s, a) \in \mathcal{U}_\mathcal{D}^*) = d^{\pi^*} \delta((s, a) \in \mathcal{U}_\mathcal{D})$. In practical situations, it is difficult to obtain the set $\mathcal{U}_\mathcal{D}$. Rather, we will be using the measure $u(s, a)$ to provide an estimate of $(s, a) \in \mathcal{U}_\mathcal{D}$. We will approximate $\delta((s, a) \in \mathcal{U}_\mathcal{D})$ as $\frac{1}{Z(s)} \exp(u(s, a))$. The exploratory policy must be the one that maximizes its visitation for the state-action pairs in $\mathcal{U}_\mathcal{D}^*$. Using $p((s, a) \in \mathcal{U}_\mathcal{D}^*)$ as a target distribution, we can obtain the optimal policy minimizing an $f$-divergence between the policy's visitation distribution and the target distribution as discussed in several previous works (Ma et al., 2022; Agarwal et al., 2024). We use Forward KL because of its performance and exploration properties. The optimization can be further simplified into a regularized RL objective.

$$
\begin{aligned}
\pi_{exp}^* &= \arg\min_\pi D_{FKL}(d^\pi(s, a) || d^*(s, a) \exp(u(s, a))) \\
&= \arg\max_\pi \mathbb{E}_{d^\pi}[u(s, a)] - D_{KL}(d^\pi(s, a) || d^*(s, a)).
\end{aligned}
\tag{2}
$$

Practically, we do not have access to $\pi^*$, preventing us from defining $\pi_{exp}^*$ exactly. We will use a biased estimate of $\pi^*$, which is the best offline policy, $\pi_D^*$. In general this isn't true, but if we restrict the marginal state visitation to be close to the dataset, i.e. ensuring $D(d^{\pi^*}(s) || d^{\pi_D^*}(s)) \leq \epsilon$, we can further simplify this objective to,

$$
\pi_{exp}^* = \arg\max_\pi \mathbb{E}_{d^\pi}[u(s, a)] - \beta D_{KL}(\pi || \pi_D^*).
\tag{3}
$$

Common offline RL algorithms (Kostrikov et al., 2021; Kumar et al., 2020) learn $\pi_D^*$ that are highly constrained to induce transitions that are found in $D$. For any given state, a policy trained using AWR (Peng et al., 2019b) has negligible probability of taking an action that is not present in the dataset, making out-of-distribution exploration difficult. We use fitted Q learning for some of our experiments to avoid the regularizations that are common to recent offline RL methods.

Note that the KL constraint on visitations has been approximated to be on behavior or policy. This approximation makes sense if the marginal state visitation is bounded to be close to the dataset. Further, it has been shown in Mao et al. (2024) that semi-gradient updates for visitation regularized offline RL is, in fact, performing the same updates as behavior regularized RL.

Since the policy $\pi_{exp}^*$ needs to be obtained only from offline data, offline RL algorithms will overly constrain $\pi_{exp}^*$ to be within the dataset, which is not ideal for an exploratory policy. Hence, we construct the exploratory policy to approximate the solution of Equation (3) directly using $\pi_D^*$, $u(s, a)$ and the offline dataset $\mathcal{D}$.

## 5.2 APPROXIMATE CONSTRUCTION

Neither online interactions nor offline RL can be used to optimize Equation (3). We use the following two sampling based approximations for $\pi_{exp}^*$:

**Single Step**: Equation (3) has a closed form solution as discussed in prior works (Peng et al., 2019a; Rafailov et al., 2024). $\pi_{exp}^*$ can be written in closed form as,

$$
\pi_{exp}^*(a|s) \propto \pi_D^* \exp\left(\frac{1}{\beta} u(s, a)\right).
\tag{4}
$$

Equation (4) provides a single step exploratory policy. The partition function can be obtained for simple discrete action spaces but for general action spaces, sampling from this distribution can be

complicated. This policy construction indicates that we can sufficiently perform exploration by balancing the uncertainty of a given dataset and the policy extracted thereof.

**Multi-Step**: Given the difficulty of sampling from a distribution such as Equation (4) in larger and continuous action spaces, we also propose an alternative way to use the uncertainty metric $u(s, a)$ for decision-making. Specifically, we formulate the following online trajectory optimization problem with a learned dynamics model $\hat{T}$ and learned reward function $\hat{r}$,

$$\pi_{exp}^*(a|s) = \arg\max_{a_0, \cdots, a_{H-1}} \mathbb{E}_{\hat{\pi}} \left[ \sum_{t=0}^{H-1} \gamma^t \left( \hat{r}(s_t, a_t) + c \cdot u(s_t, a_t) \right) + \gamma^H V_D^*(s_H) \right], \quad (5)$$

such that

$$s_0 = s, \quad a_t \sim \hat{\pi}(\cdot|s_t), \quad s_{t+1} \sim \hat{T}(s_t, a_t),$$

where $c \in (0, 1]$ is a hyperparameter that trades off between exploration and exploitation. The $H$-step rollouts are taken using $\hat{\pi}$ which a high-temperature version of the dataset policy:

$$\hat{\pi}(a|s) = \frac{e^{\pi_D^*(a|s)/T}}{\sum_{a'} e^{\pi_D^*(a'|s)/T}},$$

where $T \in \mathbb{R}^+$ is the temperature parameter of the softmax function. The argmax is over sequences of actions, but we can use model predictive control (MPC) and only commit to the first action in the sequence. This multi-step exploratory policy can be used to sample actions for arbitrary complex action spaces.

# 6 EXPERIMENTS

We present experiments that examine the exploration policy's ability to achieve the desiderata: 1) maintain closeness to the best known policy, and 2) collect data that improves the coverage of the dataset. To demonstrate these features we utilize a simple Gridworld domain and an industry-level supply chain environment we call `SupplyEnv` . The supply chain environment is backtestable with real-world data (Andaz et al., 2023; Madeka et al., 2022) and has in fact been used to train real-world policies. Through these experiments we aim to demonstrate that careful consideration of uncertainty is critical when deploying stationary exploration policies.

## 6.1 GRIDWORLD TOY DOMAIN

**Experiment Setup:** We consider a simple and canonical Gridworld environment that has four rooms where the agent must navigate from a starting location in one room to a goal location in another room. The agent receives a reward of $-1$ everywhere except the goal where it receives $0$. The agent observes its $(x, y)$ location and can take an action up, down, left, or right. We iterate between "explore" and "train" phases as would happen in the real world. The *explore* phase executes the exploratory policy in the environment to collect $\mathcal{T}$ trajectories and the *train* phase uses the combined buffer to train the policy for $K$ steps using fitted Q-learning (Ernst et al., 2005; Riedmiller, 2005) (described in Appendix D.2 and Algorithm 2). We perform $N$ such deployments. In these experiments, we show $100$ deployments with $1000$ environment and $1000$ training steps, each. To obtain an initial policy, we train a Q-function using fitted Q-iteration (Ernst et al., 2005) on a dataset of random behavior. We then calculate an uncertainty measure using (state,action)-visitation counts and deploy a stationary exploration policy (defined in Equation (6)) to collect data in the gridworld. The exploratory policy $\pi_{\exp}^*(a|s)$ samples an action as follows:

$$a \sim \exp(Q_i(s, a)) \cdot u(s, a), \quad (6)$$

where $Q_i$ is the Q-function for the $i^{\text{th}}$ iteration, $u(s, a) = \frac{1}{\sqrt{N_{s,a}}}$ (as is standard in count-based settings (Strehl & Littman, 2008; Lobel et al., 2023), and $N_{s,a}$ is the number of times action $a$ has been taken in state $s$.

**Baselines:** There are two variants of our exploration: **Single-Step** (defined in Equation (4) and Equation (6)) and **Multi-Step** (defined in Equation (5)). These are compared against prototypical exploration methodologies:

1. *Naive-$\epsilon$-greedy* chooses an uniformly random action with probability $\epsilon$ and the greedy action with probability $1 - \epsilon$.

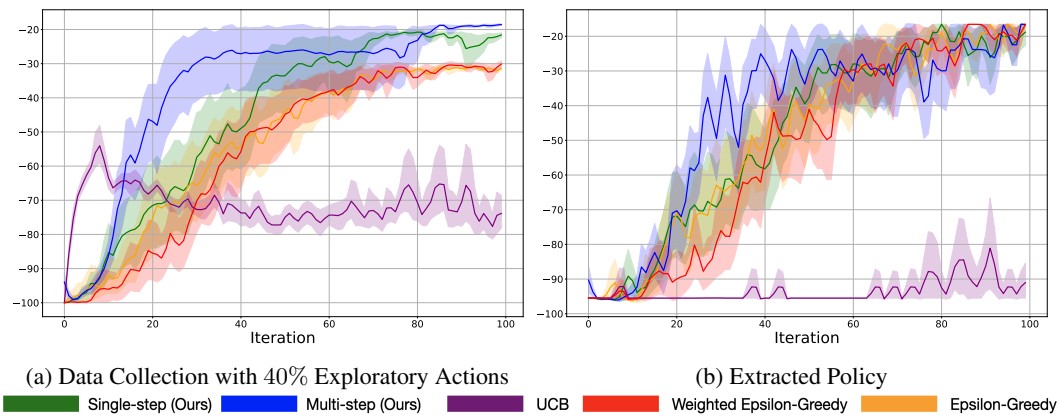

(a) Data Collection with $40\%$ Exploratory Actions  (b) Extracted Policy

██ Single-step (Ours)   ██ Multi-step (Ours)   ██ UCB   ██ Weighted Epsilon-Greedy   ██ Epsilon-Greedy

Figure 2: Comparison of our exploration policies (Single-Step and Multi-Step) against the baselines using **collect-returns** (left) and **eval-returns** (right). Each curve shows a mean of 5 seeds with a rolling average of window length 5 and the shaded region represents the 95% confidence interval.

2. ***Weighted-$\epsilon$-greedy*** chooses an action based on the uncertainty with probability $\epsilon$ and the greedy action with probability $1 - \epsilon$.

3. **Bandit-Style Reward Bonus** has no explicit exploration action, but trains a Q-network with an uncertainty augmented reward target where $r' = r(s, a) + c\frac{1}{\sqrt{N_{(s,a)}}}$, where $N(s, a)$ are tabular visitation counts, and $c \in [0, 1]$ is a fixed hyperparameter.

As a performance metric, we measure the mean return of the exploration policy and the greedy policy obtained from the offline RL algorithm and call these **eval-returns** and **collect-returns** respectively.

**Results:** Figure 2 shows the **collect-returns** (left) and **eval-returns** (right) of both of our exploratory policies compared against the baselines in the $\epsilon-$exploration setting. The results in Figure 2a demonstrate that our exploratory policies perform better when doing data collection than both variants of $\epsilon$-greedy and all outperform the UCB baseline. We suspect the UCB baseline to perform poorly because the reward targets are non-stationary and change significantly between deployments. The performance of the exploitation policies, as shown in Figure 2b, is similar, which indicates the similar quality of the samples collected during exploration. In short, our and the baseline exploration methods collect similar data but our methods do so in a safer and more performant manner.

## 6.2 APPLICATION TO A REAL-WORLD SUPPLY CHAIN

We consider an application to an inventory control problem where a retailer needs to procure inventory and balance the costs associated with over-stocking versus missing sales due to under-stocking. Madeka et al. (2022) proposed an offline RL procedure (DirectBackProp) for an inventory control problem that constructs a simple simulator using the offline data and performed model-based RL to obtain performant policies. At any given time, the state, $s$ consists of features inventory, demand, cost, time, product identifier etc appended with prior states in the history. The action, $a$ is inventory bought at any time. Andaz et al. (2023) considers learning dynamics models from offline data to relax the exogenity assumptions made by Madeka et al. (2022) on several features. Naturally, the dynamics models are only reliable on-policy (for the behavior policy) which restricts the search space for policies in the simulator. To get more performant policies, exploration is needed to add more data to improve the simulator.

**Experiment Setup:** In an ideal experiment, we would use the simulator to construct the exploratory policy, collect real-world data, add it to the existing offline dataset, improve the simulator and corresponding train a better policy. Unfortunately, this pipeline is infeasible for a conducting repeatable experiments due to (1) the streaming nature of data preventing us to repeat on the same time period and (2) costs of exploring in the real world. We rely on mimicking the various levels of fidelity between the real world and simulator with the following set up.

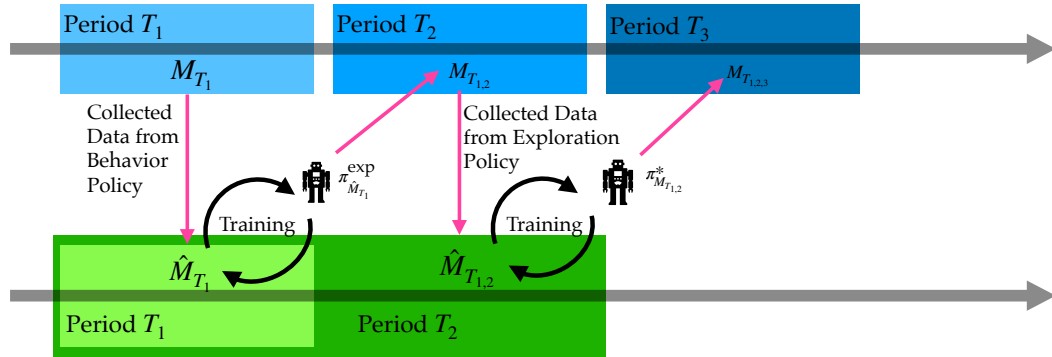

Figure 3: **Supply Chain Experiment Setup:** A low fidelity simulator $\hat{M}_{T_1}$ from the data collected on the real-world proxy $M_{T_1}$. The exploratory policy $\pi^*_{exp}$ is constructed using the simulator $\hat{M}_{T_1}$ to collect data in the next time period $T_2$. This data is added to the previous data to improve the simulator to $\hat{M}_{T_{1,2}}$. $\pi^*$ is trained on $\hat{M}_{T_{1,2}}$ and evaluated on the next time period $T_3$.

We shall represent the highest fidelity simulator, trained on all available data of the time period as $M$. It was shown that this simulator is well-calibrated with the real world (Andaz et al., 2023). This simulator will act as a proxy for the real world. Lower fidelity simulators $\hat{M}$ (which we shall call `SupplyEnv`) will be trained using data collected from $M$.

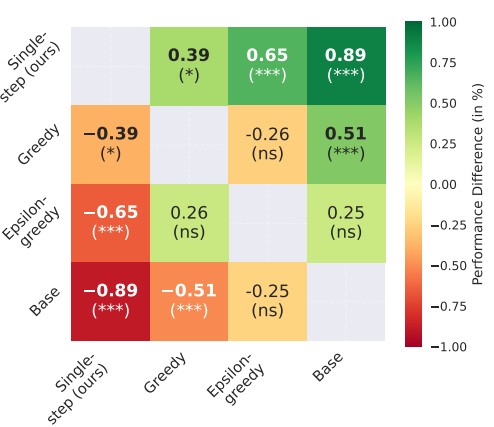

To mimic exploration in real world, we divide the timeline of dataset (5 years) into three time periods, $T_1$, $T_2$, and $T_3$. Let $M_{T_i}$ be the simulator trained using the real-world data from time segment $T_i$. $\hat{M}_{T_1}$ is constructed using data collected by a heuristic policy on $M$ for time period $T_1$. We can construct our exploratory policy $\pi^*_{exp}$ on $\hat{M}_{T_1}$. We cannot go back and collect data in $T_1$ so we deploy the exploratory policy in $M_{T_2}$ to collect data that can be added to the previously collected data to learn $\hat{M}_{T_{1,2}}$. We train a policy ($\pi^*_{M_{T_{1,2}}}$ on the improved simulator $\hat{M}_{T_{1,2}}$ using DirectBackprop and evaluate it on time period $T_3$ ($M_{T_3}$). Figure 3 describes this experiment visually.

**Constructing $\pi^*_{exp}$:**

Figure 4: This figure shows a confusion matrix describing the relative difference in performance of a policy trained using data collected by the Single-step, Greedy, and Epsilon-greedy exploration methods as well as the original policy (Base). Each box represents the percentage difference of the `row` method against the `column` method. Bold numbers denote statistical significance and "ns" denotes not statistically significant. One, two, and three asterisks denote a $p$-value of $< 0.05$, $< 0.01$, and $< 0.001$, respectively.

In the `SupplyEnv` we model uncertainty by fitting an $K$-component Gaussian Mixture Model (GMM) to a dataset of state-action pairs using a subset of the state features. We denote the resulting joint distribution as $p_{\mathcal{G}}(s_r, a)$, where $s_r = s|_{\mathcal{I}}$ is the restriction of $s$ to a selected index set $\mathcal{I} \subseteq \{1, \ldots, \dim(s)\}$. To calculate an exploratory action while in state $s$, we sample a component-reweighted version of $p_{\mathcal{G}}(a|s_r)$ shown in Algorithm 1. Due to computation restrictions, we only deal with single-step exploratory policies in `SupplyEnv`.

**Baselines:** We compare our **single-step** exploration against two common-sense exploration frameworks: 1) **greedy**, which uses $\pi^*_{\hat{M}_{T_1}}$ to explore in $M_{T_{1,2}}$, as is done in Matsushima et al. (2020), and 2) **epsilon-greedy**, which takes uniformly random action with probability $\epsilon$ and otherwise use $\pi^*_{\hat{M}_{T_1}}$.

**Results:** Figure 4 shows a comparison of the return achieved by various policies evaluated in $M_{T_3}$ in terms of their relative performance. The policies being compared are trained in three distinct

---

**Algorithm 1** Uncertainty-aware Action Resampling

---

1: **Given:**
   - Policy $\pi_M$
   - Fitted K-component GMM $p_{\mathcal{G}}(s_r, a) = \sum_i^K \alpha_i p_i(s_r, a)$
   - Temperature parameter $\tau > 0$
   - Input state $s$, with relevant feature subset $s_r = s|_{\mathcal{I}}$
2: Evaluate state-action likelihood: $p_{\mathcal{G}}(s_r, \pi_M(s))$
3: Extract *unweighted* GMM component contributions: $w_i = \alpha_i \frac{p_i(s_r, \pi_M(s))}{p_{\mathcal{G}}(s_r, \pi_M(s))}$
4: Compute reweighted components:
$$w_i' \leftarrow \frac{\exp(w_i/\tau)}{\sum_{j=1}^K \exp(w_j/\tau)} \quad \text{for } i = 1, \ldots, K$$
5: Construct reweighted GMM $p_{\mathcal{G}}' {=} \sum_i^K w_i' p_i(s_r, a)$
6: Sample new action: $a' \sim p_{\mathcal{G}}'(\cdot \mid s_r)$

---

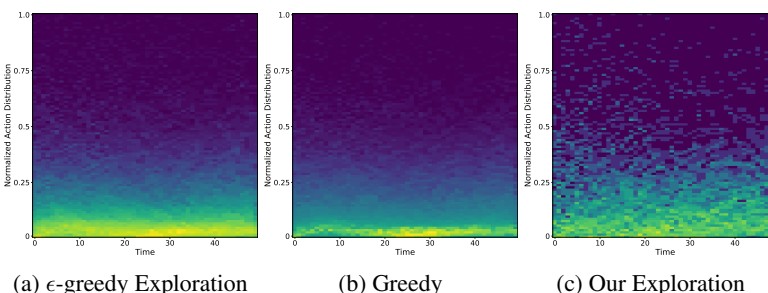

(a) $\epsilon$-greedy Exploration      (b) Greedy      (c) Our Exploration

Figure 5: This figure shows the distribution of normalized actions at each time step in the trajectory for thousands of trajectories. The bright regions denote higher density.

versions of $\hat{M}_{1,2}$, each constructed from data collected in $M_{1,2}$ by the greedy, epsilon-greedy, and our single-step exploration policies. We also compare against a policy that was only trained in $\hat{M}_1$ (Base). Firstly, we confirm that exploration is necessary by seeing that the Base policy under-performs all other methods (left column). Further, we see that the our single-step exploration method outperforms the other three policies to a statistically significant degree (top row). Importantly, the greedy policy does not sufficiently cover the state space to learn the most robust policy in the next round of deployment. The action distributions over time shown in Figure 5 give us a qualitative example of what the exploration policies are doing during deployment. When comparing the $\epsilon$-greedy exploration action distribution (Figure 5a) to that of the greedy exploration (Figure 5b), we can see that the former is simply a uniformly higher variance version of the latter. In contrast, our exploration (Figure 5c) demonstrates a much less naively higher variance.

## 7 CONCLUSION

In many real-world settings, policies cannot be continuously updated online; we proposed exploration strategies for such domains. We adopted deployment efficiency as a relevant evaluation metric, and showed how our algorithms lead to high return and deployment efficiency. We constructed two exploration policies (single-step and multi-step) by combining a model of the uncertainty of a dataset along with the optimal policy that can be extracted from it. Along with theoretical bounds on the sub-optimality of these policies, we provide extensive experiments in the Nav-Chambers environment and a well-calibrated simulator of a real-world supply chain environment. Our proposed methods balance safety with exploration and show that it is possible to maintain performance while exploring in the deployment efficient regime.

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

APPENDIX

## A  OFFLINE RL AND PESSIMISM

Offline RL refers to the class of RL algorithms that train on a fixed dataset. Due to the inherent overestimation biases (Hasselt, 2010) in RL algorithms, naively trained policies have a tendency to steer out of the dataset towards the highly-biased value estimates. As a result, regularizations are added to the ensure pessimism in the value functions. These regularizations can be behavior based (Kumar et al., 2019; Wu et al., 2019), value based (Kumar et al., 2020), visitation based (Sikchi et al., 2023) or reward based (Kidambi et al., 2021; Yu et al., 2020). Recent works have shown several interconnections among these (Sikchi et al., 2023; Mao et al., 2024). We consider the following constraint optimization as offline RL optimization problem:

$$\max_{\pi} \quad J(\pi, \mathcal{M})$$
$$s.t. \quad D(p^{\pi}(s, a) || \hat{p}(s, a)) \leq \epsilon, \tag{7}$$

Equation 7 has been studied through common offline RL and imitation learning algorithms (Nachum et al., 2019; Sikchi et al., 2023). We approximate the optimization problem using a pessimistic MDP (as defined in Kidambi et al. (2021)). The pessimistic MDP ensures that $D(p^{\pi}(s, a) || \hat{p}(s, a)) \leq \epsilon$ is always satisfied as any transition outside the dataset is heavily penalized.

### A.1  PROOF OF THEOREM 5.4

We will be introducing some definitions and lemmas in order to prove Theorem 5.4. We begin by defining hitting time,

**Definition A.1.** *(Hitting time)* Given an MDP $\mathcal{M}$, starting state distribution $\rho_0$, state-action pair $(s, a)$ and a policy $\pi$, the hitting time $T_{(s,a)}^{\pi}$ is defined as the random variable denoting the first time action $a$ is taken at state $s$ by $\pi$ on $\mathcal{M}$, and is equal to $\infty$ if $a$ is never taken by $\pi$ from state $s$. For a set of state-action pairs $S \subseteq S \times A$, we define $T_S^{\pi} \stackrel{\text{def}}{=} \min_{(s,a) \in S} T_{(s,a)}^{\pi}$. In other words, it is the time it takes to arrive at state $s$.

Using the definition, we can introduce the following Lemma that bounds the returns obtained in the pessimistic MDP $\mathcal{M}_{\mathcal{D}}$ using the returns in the true MDP $\mathcal{M}$.

**Lemma A.2.** *Let $\mathcal{M}$ be a given MDP. The following is true for any offline dataset $\mathcal{D}$.*

$$J(\pi, \mathcal{M}) - \frac{2R_{max}}{1 - \gamma} \mathbb{E}[\gamma^{T_{\mathcal{U}_{\mathcal{D}}}^{\pi}}] \leq J(\pi, \mathcal{M}_{\mathcal{D}}) \leq J(\pi, \mathcal{M}) \tag{8}$$

*Proof.* A more extensive theorem and corresponding proof exists in Kidambi et al. (2021). The definition of pessimistic-MDP in the two works is slightly different leading to different bounds.

The rollout of any policy $\pi$ on the pessimistic MDP $\mathcal{M}_{\mathcal{D}}$ would be the same as that of the true MDP $\mathcal{M}$ as long as an unknown state ($s \in \mathcal{U}_{\mathcal{D}}$) is encountered. At that point, the return of the policy on the pessimistic MDP will be $\frac{-R_{max}}{1 - \gamma}$. The maximum return of the policy for that rollout segment can be $\frac{R_{max}}{1 - \gamma}$. Hence,

$$J(\pi, \mathcal{M}) - \frac{2R_{max}}{1 - \gamma} \mathbb{E}[\gamma^{T_{\mathcal{U}_{\mathcal{D}}}^{\pi}}] \leq J(\pi, \mathcal{M}_{\mathcal{D}}) \tag{9}$$

Since $\mathcal{M}_{\mathcal{D}}$ is a pessimistic MDP for the corresponding MDP $\mathcal{M}$ with each state in $\mathcal{M}_{\mathcal{D}}$ having rewards less than or equal to $\mathcal{M}$, for any policy, the return of will be higher in $\mathcal{M}$ compared to $\mathcal{M}_{\mathcal{D}}$ i.e. $J(\pi, \mathcal{M}_{\mathcal{D}}) \leq J(\pi, \mathcal{M})$. □

With this bound, we can compare the returns obtained by policies trained on the pessimistic MDP with the corresponding returns in true MDP. To simplify this comparison, we use the following Lemma relates hitting times to visitation distributions.

**Lemma A.3.** *(Kidambi et al., 2021) (Hitting time and visitation distributions) For any set $\mathcal{X} \subseteq \mathcal{S} \times \mathcal{A}$, and any policy $\pi$, we have $\mathbb{E}[\gamma^{T_{\mathcal{X}}^{\pi}}] \leq \frac{1}{1-\gamma} d^{\pi}(\mathcal{X})$.*

Now, we have all the components required to prove Theorem 5.4.

**Theorem 5.4.** *Let $\mathcal{M}$ be a given MDP and $\mathcal{D}$ be the offline dataset. Let $\pi^* = \arg\max_{\pi} J(\pi, \mathcal{M})$ be the true optimal policy on $\mathcal{M}$ and $\pi_D^* = \arg\max_{\pi}$ Equation (1) $\approx \arg\max_{\pi} J(\pi, \mathcal{M}_{\mathcal{D}})$ be the best offline RL policy on dataset $\mathcal{D}$. The following holds: $J(\pi^*, \mathcal{M}) - J(\pi_D^*, \mathcal{M}) \leq \frac{2R_{max}}{(1-\gamma)^2} d^{\pi^*}(\mathcal{U}_{\mathcal{D}})$.*

*Proof.* From Lemma A.2,

$$J(\pi, \mathcal{M}) - \frac{2R_{\max}}{1-\gamma}\mathbb{E}[\gamma^{T_{\mathcal{U}_{\mathcal{D}}}^{\pi}}] \leq J(\pi, \mathcal{M}_{\mathcal{D}}) \leq J(\pi, \mathcal{M})$$

$$J(\pi^*, \mathcal{M}) - J(\pi^*, \mathcal{M}_{\mathcal{D}}) \leq \frac{2R_{\max}}{1-\gamma}\mathbb{E}[\gamma^{T_{\mathcal{U}_{\mathcal{D}}}^{\pi^*}}]$$

From Lemma A.3, we replace the upper bound

$$J(\pi^*, \mathcal{M}) - J(\pi^*, \mathcal{M}_{\mathcal{D}}) \leq \frac{2R_{\max}}{(1-\gamma)^2} d^{\pi}(\mathcal{U}_{\mathcal{D}})$$

By definition, we know that $J(\pi^*, \mathcal{M}_{\mathcal{D}}) \leq J(\pi_{\mathcal{D}}^*, \mathcal{M}_{\mathcal{D}}) \leq J(\pi_{\mathcal{D}}^*, \mathcal{M})$. Thus we substitute

$$J(\pi^*, \mathcal{M}) - J(\pi_{\mathcal{D}}^*, \mathcal{M}) \leq \frac{2R_{\max}}{(1-\gamma)^2} d^{\pi}(\mathcal{U}_{\mathcal{D}})$$

$\square$

# B    CONNECTIONS TO COMMONLY USED EXPLORATION

As discussed above, defining $u(s, a)$ is a major design decision that needs to be taken and there is no optimal way to define uncertainty. We can show that different heuristic definitions of $u(s, a)$ and $\pi_D^*$ can draw connections to commonly used exploration techniques.

$\epsilon$-**greedy**: Suppose $u(s, a)$ is not learned but defined using a crude heuristic: For any state $s$, with probability $1 - \epsilon$, the action is not uncertain i.e. $\exp\frac{1}{\beta}u(s, a) = 1$ and with probability $\epsilon$, the action is uncertain with inversely depending on $\pi_D^*$ i.e. $\exp\frac{1}{\beta}u(s, a) = \frac{c}{\pi_D^*(a|s)}$. Then, the corresponding exploration turns out to be $\epsilon$-greedy. What does the uncertainty mean here? The value $\frac{c}{\pi_D^*(a|s)}$ means that the uncertainty is lower for policy actions which makes some sense as the actions taken by the policy are already known and present in the dataset.

**Bandit-style UCB**: Bandit-style UCB produces a policy that takes action according to $Q(s, a) + u(s, a)$ where $u(s, a)$ is of a specific function depending on the frequency of actions. UCB is very efficient exploration method as it provides a logarithmic regret bound in bandits. If $\pi_D^*$ is assumed to be a softmax of the offline Q function i.e. $\pi_D^* \propto \exp Q(s, a)$, the exploration policy $\pi_{exp}^*$ becomes a soft version of the Bandit-style UCB. It must be noted though, for most offline RL algorithms, $\pi_D^*$ is not $\propto \exp Q(s, a)$ (Kostrikov et al., 2021; Sikchi et al., 2023).

**Uncertainty aware reward bonus**: A common way of exploring in RL is to use uncertainty based reward bonus. This method is inspired from UCB and is formalized better in Strehl & Littman (2008). Here the Q function are trained to be optimistic by adding $u(s, a)$ to the reward function. Mathematically, $Q^{\pi}(s, a) = \mathbb{E}_{\pi}[\sum_t \gamma^t(r(s_t, a_t) + c \cdot u(s_t, a_t))]$. These algorithms are online algorithms converging to optimal Q functions as $u(s, a) \to 0$.

# C    ADDITIONAL RESULTS

We ablate over different values of epsilon to demonstrate the importance of uncertainty across a variety of exploration budgets. Figure 6 shows the **eval-returns** (bottom row) and **collect-returns** (top row) of our method and various baselines across different epsilons. These results show the $\epsilon$-exploration setting. Finally, from left to right, the value of $\epsilon$ increases. We see relative performance improvements on the **collect-returns** of our single- and multi-step methods as $\epsilon$ increases.

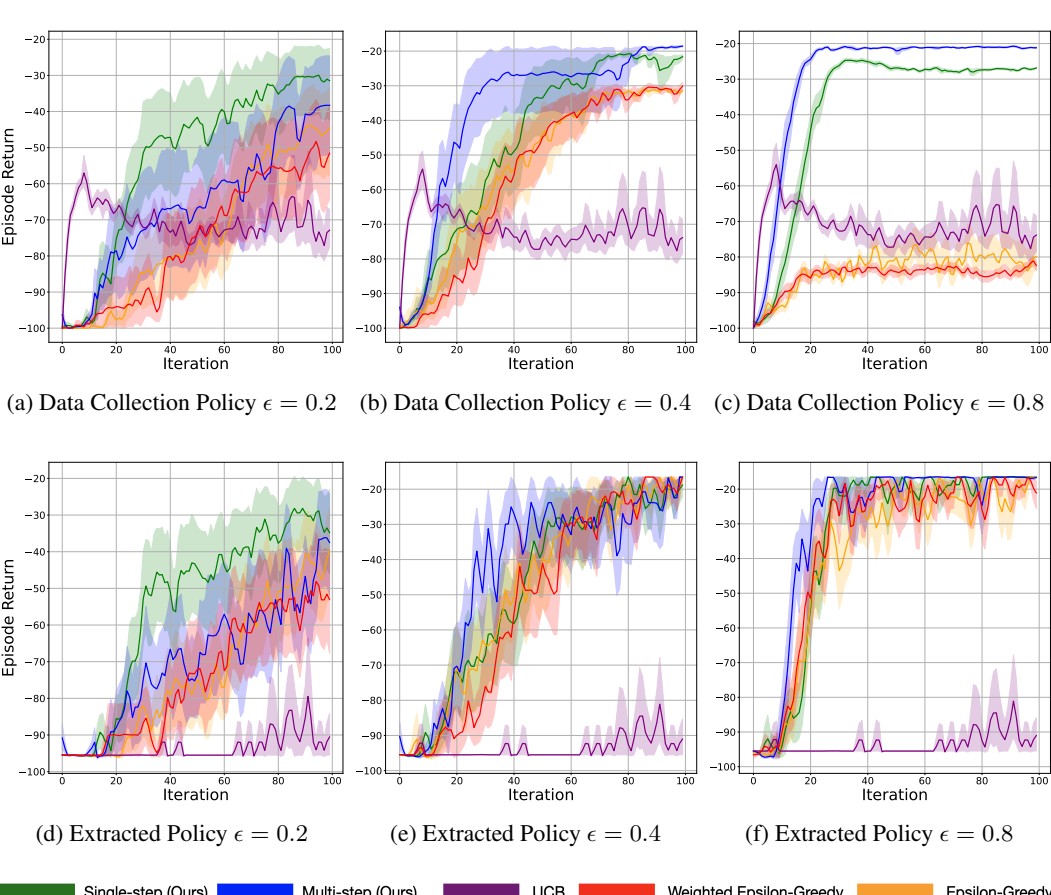

Figure 6: Comparison of our exploration policies (Single-Step and Multi-Step) against the baselines using **collect-returns** (top row) and **eval-returns** (bottom row). Each curve shows a mean of 5 seeds with a rolling average of window length 5 and the shaded region represents the 95% confidence interval. From left to right, each column represents a different value of epsilon ($\epsilon$ used for the $\epsilon$-exploration across methods.)

# D   EXPERIMENTAL SETUP

## D.1   SUPPLY CHAIN ENVIRONMENT

We use the supply chain environment described in detail in (Madeka et al., 2022; Andaz et al., 2023). Here the RL agent makes decisions about buying items from vendors. The agent takes in a state $x$ and its history of length $H$ to predict the next action. The state consists of several features including demand, inventory, cost, time of year, etc. The authors of Madeka et al. (2022) introduced an algorithm to solve this domain and use real-world data by assuming the environmental setup to be an Exogenous-MDP. A simulator is created with the learned dynamics models for the endogenous components and the exogenous components are sampled from the real-world data. Because of the exogenous assumption, this simulator allows for back-testing of the policies and is calibrated with the real world. An RL agent is trained in the simulator using the DirectBackprop algorithm (Madeka et al., 2022).

While our method is designed for real-world exploration, it is infeasible to test in the real world due to time-constraints (each step in the simulator represents one week). To overcome this, we partition our data into three segments, $T_1, T_2$, and $T_3$ that represent train, explore and test periods, respectively. Further, we construct a reduced fidelity versions of the data by clamping some of the extreme (but highly rewarding) features; this allows us to generate simulators that mimics low data coverage.

## D.2   NAV-CHAMBER EXPERIMENTS

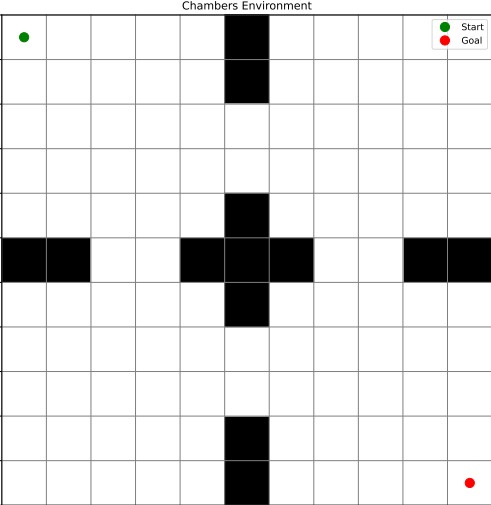

Figure 7: Obstacle map for the Nav-Chambers environment.

The Nav-Chambers is a gridworld-like environment where the action space is [Up, Down, Left, Right] and the reward is $-1$ everywhere except the goal, where it is $+1$. In the Nav-Chambers environment, we implement deployment efficient exploration using count-based uncertainty estimation and train extract a policy using fitted Q-iteration, as described in Algorithm 2. The results shown use 100 deployments, each with 1000 train steps and 1000 collect steps.

---

**Algorithm 2** Deployment Efficient Exploration with Fitted Q-Iteration

---

1: Initialize dataset of interactions $\mathcal{D} \leftarrow \{\}$, initialize Q-function $Q_\theta(s, a)$
2: **for** $n = 1$ to $N$ **do**
3:      **for** $k = 1$ to $K$ **do**
4:          Sample $s$ from environment or replay buffer
5:          Sample action $a \sim \pi^{\text{exploration}}(a \mid s)$
6:          Execute $a$, observe $r$, $s'$
7:          Add transition to dataset: $\mathcal{D} \leftarrow \mathcal{D} \cup \{(s, a, r, s')\}$
8:      **end for**
9:      Fit $Q_\theta$ to minimize Bellman loss on $\mathcal{D}$:
$$\mathcal{L}(\theta) = \sum_{(s,a,r,s') \in \mathcal{D}} \left( Q_\theta(s, a) - \left[ r + \gamma \max_{a'} Q_{\theta'}(s', a') \right] \right)^2$$
10:      Optionally update target network $\theta' \leftarrow \theta$
11: **end for**
12: **return** final Q-function $Q_\theta$

---

Across the experiments, the exploration policy is deployed a total of 100 times, collecting 1000 transitions and training the policy for 1000 gradient steps. After each deployment, we use fitted

# E    Uncertainty-aware Exploration Policy in Supply Chain Environment

Because the Supply Chain Environment has high dimensional state and action spaces, we approximate uncertainty with a $N$-component Gaussian mixture model (GMM), and then construct an uncertainty-aware exploration policy by reweighting the model components at explore-time. The Gaussian mixture model $p_\mathcal{G}$ is learned to approximate the joint distribution of $s_r$ and $a$ present in the dataset, where $s_r$ is defined in Section 6.2. To take an uncertainty-maximizing action in the Supply Chain Environment, the agent first calculates its deterministic nominal action $a = \pi(s)$. Next, calculate the likelihood of the proposed state-action pair with respect to $p_\mathcal{G}$ and determine contributions $\mathbf{w}$ of each of the $N$ component Gaussians. Lastly, resample the action $a$ from the conditional GMM $p'_\mathcal{G}(a|s_r)$ but reweigh the components using a higher temperature distribution based on the component contributions $\mathbf{w}$ where
$$w'_i = \frac{\exp(w_i/\tau)}{\sum_i \exp(w_i/\tau)}. \tag{10}$$
The procedure is outlined in detail in Algorithm 1.

# F    Exploration Trajectories in the Supply Chain Environment

In the Supply Chain Environment, we have compared three different exploration strategies in Section 6.2. We have observed that our exploration policy does better than greedy and $\epsilon$-greedy exploration policies. While it does makes sense as our strategy is theoretically better than the naive exploration strategies. But we investigate how different these explorations are qualitatively. It is not trivial to directly visualize the policies in the Supply Chain Environment. We plot the distribution of actions taken for every timestep of the trajectory to visualize the exploratory policies. Figure 5 shows these plots of four policies: a heuristic policy (one that takes actions to maintain static inventory levels), a greedy policy, an $\epsilon$-greedy policy and our exploratory policy. It can be seen that while the heuristic policy is totally random across the entire space, the $\epsilon$-greedy stays very close to the greedy policy. Our exploration on the other hand is not as random (and unsafe) as the heuristic policy but still explores farther than the $\epsilon$-greedy.

