# OpenReview forum: "Exploration for Deployment-Efficient Reinforcement Learning Agents"
_ICLR.cc/2026/Conference — Submitted to ICLR 2026_

### Official Review · Reviewer_MGgf · 2025-10-29

**Soundness:** 1
**Presentation:** 2
**Contribution:** 1
**Rating:** 2
**Confidence:** 3

**Summary:**

This paper focuses on offline-to-online RL by designing a stationary policy to improve exploration. This paper proposes an exploration policy that aims to reduce the uncertainty in the offline data while staying within the known safe regions. Empirical results on the gridworld and supply chain environment are reported.

**Strengths:**

(+) The paper is written clearly.

**Weaknesses:**

(-) The analysis in this paper is not well justified, e.g., Assumption 5.3.

(-) The experiment section lacks standard baseline methods. Only naive baselines are compared.

**Questions:**

Q1. How is the uncertainty $u$ computed in Def 5.1?

Q2. In Line 208, what are the references for the following statement: "This assumption is also made by most offline RL algorithms using Equation (1) as their objective"?

Q3. Can you provide a comparison with standard offline RL algorithms?

---

> ### Author Response · Authors · 2025-12-03
> **Author response**
>
> > How is the uncertainty computed in Def 5.1?
>
> Our work focuses on how to soundly incorporate the coverage of a dataset into an exploration policy. We do not fix our method to one notion of uncertainty, as we have found it to be domain dependent. For instance, in discrete state-action spaces (e.g., our gridworld domain), it is trivial to compute visitations. However, in a domain like the supply chain environment, where most of the dynamics are exogenous and only the inventory can be directly affected by the actions, it is sufficient to use a gaussian mixture model over the action trajectories. It is possible to fit energy-based models to a dataset, as an alternative uncertainty modeling scheme. Regardless, there are many choices one can make on modeling uncertainty such as [1,2].
>
> > In Line 208, what are the references for the following statement: "This assumption is also made by most offline RL algorithms using Equation (1) as their objective"?
>
> The goal of offline RL is to ensure that the policy stays within the data distribution. There have been two ways that methods have used to ensure this: behavior regularization and visitation regularization. Methods using behavior regularization (such as IQL [3]) ensure that the policy distribution stays within the dataset which implies visitation regularization. On the other hand, methods like  [4] directly optimize over the visitation regularization.
>
> > Can you provide a comparison with standard offline RL algorithms?
>
> Standard offline algorithms do not perform any exploration. For each of our settings, we have chosen the **best** offline algorithm for that setting as shown by [5].
>
> [1] Janner, Michael, et al. "When to trust your model: Model-based policy optimization." Advances in neural information processing systems 32 (2019).
> [2] Luis, C. E., Bottero, A. G., Vinogradska, J., Berkenkamp, F., & Peters, J. (2023, April). Model-based uncertainty in value functions. In International Conference on Artificial Intelligence and Statistics (pp. 8029-8052). PMLR.
> [3] Kostrikov, Ilya, Ashvin Nair, and Sergey Levine. "Offline Reinforcement Learning with Implicit Q-Learning." International Conference on Learning Representations.
> [4] Sikchi, Harshit, et al. "Dual RL: Unification and New Methods for Reinforcement and Imitation Learning." The Twelfth International Conference on Learning Representations.
> [5] Madeka, Dhruv, et al. "Deep inventory management." arXiv preprint arXiv:2210.03137 (2022).

---

### Official Review · Reviewer_XRiF · 2025-11-01

**Soundness:** 2
**Presentation:** 2
**Contribution:** 2
**Rating:** 4
**Confidence:** 3

**Summary:**

This paper studies deployment efficient reinforcement learning, where the number of distinct data collection policies is relatively low compared to the number of updates to the policy. The authors argue that safely improving a dataset requires a deployment efficient algorithm with a carefully constructed data collection policy. They introduce a framework with a stationary exploration policy that aims to reduce out-of-distribution uncertainty while maintaining strong returns. They establish theoretical guarantees of this exploration framework without finetuning and demonstrate their method on a large-scale supply chain environment with real-world data.

**Strengths:**

1. The setting of deployment efficient RL is important.
2. The paper is technically solid, the proof looks correct.
3. There are experimental results supporting theoretical results.

**Weaknesses:**

1. The setting is not clear and seems restrictive. It is not clear whether the setting is tabular or continuous. According to the definition of $u(s,a)$, it seems that the state-action space must be discrete, which is very restrictive in real-world applications. Is it possible to extend the method to more general function approximation? How is $u(s,a)$ rigorously defined there?

2. Given Assumption 5.3, Theorem 5.4 seems to be a straightforward result. Bounding the difference between the real MDP and a pessimistic absorbing MDP is a standard approach in various previous works.

3. The selection of the policy $\pi_{exp}^\star$ is according to the standard approach of maximizing the reward (here is the uncertainty measure $u$) with a KL constraint. Could you please explain the novelty here?

**Questions:**

Please see the weakness

---

> ### Author Response · Authors · 2025-12-03
> **Author response**
>
> > Given Assumption 5.3, Theorem 5.4 seems to be a straightforward result.
>
> Assumption 5.3 is the convergence guarantee for offline RL algorithms and their respective optimization problems. Theorem 5.4 combines this assumption (that the offline RL algorithms converged) to the properties of the pessimistic MDP to produce the suboptimality bound.
>
> > It is not clear whether the setting is tabular or continuous , is it possible to extend the method to more general function approximation?
>
> Thank you for your question. We use function approximation for all our policies, and only use a tabular method in calculating the uncertainty for the gridworld environment. To be clear, the gridworld environment has a discrete state and action space while the supply chain environment has a large, and continuous state space with a unidimensional continuous action space.
>
> > The selection of the policy is according to the standard approach of maximizing the reward (here is the uncertainty measure ) with a KL constraint. Could you please explain the novelty here?
>
> The novelty in this formulation is the reversal of commonly held beliefs on learning from static datasets. The standard approach is to maximize reward using a KL constraint to the target dataset (or policy). While this is beneficial for learning safe, expected behavior, it is not ideal (or even designed)  for exploration. Our work explicitly tackles the sub-optimiality of the dataset by attempting to balance exploration in known regions (the KL constraint) with unknown regions (the uncertainty reward). This formulation explicitly addresses the exploration challenges inherent with conservative offline RL methods.

---

### Official Review · Reviewer_TFxf · 2025-11-03

**Soundness:** 2
**Presentation:** 2
**Contribution:** 2
**Rating:** 4
**Confidence:** 4

**Summary:**

The paper studies deployment-efficient exploration for offline-to-real RL: when continual online updates are infeasible, deploy a small number of stationary exploration policies to enrich data, then retrain offline. It introduces an uncertainty-weighted, KL-regularized explorer and an MPC variant that plans with a shaped reward  A theoretical bound links performance gaps to optimal-policy visitation of dataset “unknowns,” motivating targeted exploration. Experiments on a toy gridworld and a supply-chain simulator suggest better data collection (and comparable evaluation returns) than simple ϵ-greedy/UCB baselines under limited deployments.

**Strengths:**

The formalization is reasonable, but the main bound (Theorem 5.4) hinges on strong assumptions (offline optimizer finds the optimal policy for the pessimistic MDP; the KL on visitation replaced by a KL on actions), and the experiments do not fully validate the conditions under which the framework is guaranteed to help. The supply-chain evaluation uses an indirect, multi-simulator protocol with heuristic uncertainty and limited ablations.

**Weaknesses:**

1. Assumptions behind the bound are strong and under-tested.
Theorem 5.4 assumes the offline learner finds the optimal policy for the pessimistic MDP and then replaces a visitation-level divergence with a policy-level KL. The paper does not empirically probe when these surrogates are tight (e.g., via measuring actual state-visitation drift vs. action KL), nor does it test sensitivity of performance to that approximation.
2. Baselines are weak for the stated setting.
In the deployment-constrained regime, there exist stronger comparators than ϵ-greedy/naive UCB: deployment-efficient MBO (e.g., model-based data collection à la prior work cited), conservative online fine-tuning with verification/shielding, uncertainty-aware behavior cloning with data-selection, or “collect-once” behavior-regularized explorers. The paper cites several lines but does not instantiate competitive versions under the same deployment budget.

**Questions:**

Metricization of deployment efficiency.
Can you report: (a) number of distinct deployed explorers, (b) trajectories per deployment, (c) per-deployment collect-return and regret, and (d) any safety/constraint violation proxies during exploration?

---

> ### Author Response · Authors · 2025-12-03
> **Author response**
>
> > Theorem 5.4 assumes the offline learner finds the optimal policy for the pessimistic MDP and then replaces a visitation-level divergence with a policy-level KL. The paper does not empirically probe when these surrogates are tight
>
> As long as the states visited by the policy $d^\pi(s)$ is close to reference ($d^{\pi^*}(s)$), the regularization is the same. We would also like to point out that it is known in the literature [1] that most algorithms using visitation regularization are in principle still optimizing over the regularization of the policy.
>
> [1]: Mao, Liyuan, et al. "ODICE: Revealing the Mystery of Distribution Correction Estimation via Orthogonal-gradient Update." The Twelfth International Conference on Learning Representations.
>
> > Baselines are weak for the stated setting.
>
>  In the real world, any exploration method that updates their policy distributions (either through updates of the policy network or through other parameters like beliefs and uncertainties) while executing is unsafe and cannot be verified for execution in the real world. The majority of exploration methods and offline-to-online algorithms cannot be used in our setting. Policy distributions can only be trained offline and need to be verified before the agent executes them.
>
> > Can you report: (a) number of distinct deployed explorers, (b) trajectories per deployment, (c) per-deployment collect-return and regret, and (d) any safety/constraint violation proxies during exploration?
>
> Safety violations: In both environment, we are optimizing for reward with no notion of hard safety constraints.
> Per deployment regret: this is an important metric to report, but we regretfully are not able to provide these results at this time.
> **Gridworld Experiments**:
> - Number distinct deployed explorers: 100
> - Trajectories / deployment: 10
>
> **SupplyEnv Experiments:**
> - Number distinct deployed explorers: 1
> - Trajectories / deployment: >10,000

---

### Official Review · Reviewer_6xLW · 2025-11-03

**Soundness:** 2
**Presentation:** 3
**Contribution:** 3
**Rating:** 4
**Confidence:** 3

**Summary:**

The paper proposes a deployment-efficient exploration framework for offline-to-online reinforcement learning in safety- or cost-sensitive domains. It defines dataset suboptimality via visits to uncertain state-action pairs, derives a bound linking missing coverage to return loss, and constructs stationary single- and multi-step exploratory policies that stay near the dataset policy while targeting uncertain regions, with theory and experiments to validate effectiveness.

**Strengths:**

1. The paper gives a clear suboptimality characterization: $J\left(\pi^{\star}, M\right)-J\left(\pi_D^{\star}, M\right) \leq \frac{2 R_{\max }}{(1-\gamma)^2} d_{\pi^*}\left(U_D\right)$, which cleanly isolates "missing states of the optimal policy" as the only thing exploration needs to fix. This is a useful, actionable target for data-collection policies.

2. The exploratory policy is derived from a principled divergence-regularized objective so exploration is explicitly balanced against staying close to a verified baseline policy - exactly what deployment-efficient RL needs.

3. The paper provides two constructive approximations (single-step exponential reweighting and multistep MPC-style planning with an uncertainty bonus) that show the abstract objective can be instantiated in both discrete and continuous/large action spaces without online policy finetuning.

**Weaknesses:**

1. The key quantity $u(s, a)$ ("likelihood of being outside the dataset") is only heuristically estimated from counts/GMMs; the guarantees rely on it tracking the true unknown set $U_D$, but the paper does not give an error-to-performance translation for misspecified $u(s, a)$.

2. The bound and the construction assume the offline policy $\pi_D^{\star}$ is already optimal on the pessimistic MDP $M_D$; this is a strong assumption that pushes difficulty into the offline learner and is not relaxed in the main theorem.

3. The multi-step exploratory policy requires a learned dynamics model $\hat{T}$ and reward $\hat{r}$; the method does not analyze model-bias accumulation in the planning rollout, so it is unclear how far from the dataset support the MPC variant can safely explore.

**Questions:**

1. The objective $\max_\pi \mathbb{E}_{d^*}[u(s, a)]-\beta D_{\mathrm{KL}}(\pi \| \pi_D^*})$ is motivated via a visitation-level KL; can the authors formalize when the policy-level KL is no longer a good surrogate (e.g., when dataset-induced state marginals differ a lot)?

2. The suboptimality bound depends on $d_{\pi^*}\left(U_D\right)$, which is not observable. Is there a practical upper bound in terms of the learned $u(s, a)$ that could be monitored to decide when to stop deploying the exploratory policy?

3. For the multi-step version (Eq. (5)), how sensitive is the exploration target to model errors in $\hat{T}$ near the boundary of the dataset support, and can a conservative backup (e.g., HALT transition) be added without breaking improvement?

4. The framework fixes $\pi_{\exp }$ during a deployment window to satisfy verification constraints. Could the analysis be extended to a piecewise-stationary schedule and still retain the same form of the suboptimality reduction bound?

---

> ### Author Response · Authors · 2025-12-03
> **Author response**
>
> > The key quantity ("likelihood of being outside the dataset") is only heuristically estimated from counts/GMMs; the guarantees rely on it tracking the true unknown set
>
> The aim of our work is to provide the framework for real-world exploration, and not to prescribe the practical definitions of u(s, a). We define u(s, a) using the theoretically ideal measure of uncertainty as representing the likelihood of being outside the dataset. This theoretically ideal definition is not achievable in practice for most domains. In our experiments, we use the best measures for uncertainty according to the respective domains.
>
> > offline policy is already optimal on the pessimistic MDP ; this is a strong assumption
>
> We agree that not all offline RL algorithms will assuredly lead to the optimal policy on the pessimistic MDP. As such, we do not suggest any one particular algorithm to do this; even in our experiments we opt to use a DQN for the gridworld and the Direct Backprop [1] algorithm. The Direct Backprop algorithm was shown to produce the best performing policies using data from a supply chain environment very similar to ours [1]. Again, the aim of our work is to provide the framework for real-world exploration, and not to prescribe the practical definitions.
>
> > The multi-step exploratory policy requires a learned dynamics model and reward; the method does not analyze model-bias accumulation in the planning rollout, so it is unclear how far from the dataset support the MPC variant can safely explore.
>
> In the real world, you cannot explore far away from the dataset support. This is because all the network distributions become unpredictable outside the distribution they are trained on and such networks cannot be executed safely in the real world. The exploration strategy that we propose is to explore the region of the state space that is outside but very close to the data distribution.
>
> > when the policy-level KL is no longer a good surrogate (e.g., when dataset-induced state marginals differ a lot)?
>
> Due to our real world setup it is ensured that the two marginals are not different. We have mentioned that policies need to be verified before execution in the real world. A policy that has deviated from the known data distribution significantly will not be safe. In the real world, the only possible exploration can be on the very edges of the data distribution.
>
> > Is there a practical upper bound in terms of the learned  that could be monitored to decide when to stop deploying the exploratory policy?
>
> Executing the exploratory policy is safe (it is verified before executing in the real world). Practically, the exploratory policy executes to collect more data but is more costlier than the greedy policy. So, the tradeoff is between the executing costs and collecting more data.
>
> > For the multi-step version (Eq. (5)), how sensitive is the exploration target to model errors in near the boundary of the dataset support,
>
> This is a good question. Our results do not say anything about the sensitivity of long-horizon errors. We believe this is not a trivial question
>
> > Could the analysis be extended to a piecewise-stationary schedule and still retain the same form of the suboptimality reduction bound?
>
> This is a great suggestion. We believe that the answer to this question is yes, but we do not have any theoretical results to support this. In fact, in our initial experiments on the gridworld we tested a similar idea where the exploration would only begin after K steps (sampled uniformly over the expected horizon length). We found that this did not significantly impact the results, so decided to omit them from the main paper.
>
>
> [1]: Madeka, Dhruv, et al. "Deep inventory management." arXiv preprint arXiv:2210.03137 (2022).

---

### Official Review · Reviewer_LM7e · 2025-11-06

**Soundness:** 1
**Presentation:** 2
**Contribution:** 2
**Rating:** 2
**Confidence:** 3

**Summary:**

This paper proposes a framework for deployment-efficient exploration in reinforcement learning—motivated by real-world constraints where policy updates require verification and redeployment is costly. The setting lies between offline and online RL: the agent can collect additional data through limited, pre-approved deployments but cannot update its policy mid-deployment.

The authors suggest learning an offline conservative policy, then constructing an exploration policy that tilts the behavior distribution toward uncertain regions, $\pi_{\text{exp}}(a|s) \propto \pi_D(a|s)\exp(u(s,a)/\beta)$, where u(s,a) is an uncertainty function. The idea is to collect “useful but safe” new data for later offline retraining. They present several theoretical results linking coverage to sub-optimality and conduct experiments in a Gridworld domain and a supply-chain simulator.

**Strengths:**

The problem setting—safe data collection under limited deployment budgets—is realistic and practically relevant, especially for industrial RL applications. The high-level motivation of optimizing exploration to improve future offline training is a good idea, though this is well studied in the area of reward-free RL (e.g. Jin et al. 2020, Wang et al. 2020, Chen et al. 2022, Wagenmaker et al. 2022, Amortila et al. 2024). The inclusion of "real-world" experiments in a supply-chain simulator demonstrate some potential, though the method used in Section 6.2 does not seem related to the methods proposed in the rest of the paper.

Citations:
Jin et al. 2020: https://arxiv.org/pdf/2002.02794
Wang et al. 2020: https://arxiv.org/pdf/2006.11274
Chen et al. 2022: https://arxiv.org/pdf/2206.10770
Wagenmaker et al. 2022: https://arxiv.org/pdf/2201.11206
Amortila et al. 2024: https://arxiv.org/pdf/2403.06571

**Weaknesses:**

**Theoretical novelty and correctness**

The theoretical developments seem to mostly build on Kidambi et al. 2021 (including the main theoretical bound), but somehow without its formality or correctness. The new developments are sloppy and in parts incorrect. Many equalities are handwaved completely and/or incorrect as stated (e.g. Equation 1: \hat{p} is undefined, not specified which (s,a) is being considered), many approximate equalities $\approx$ that are informal and unclear). The main theoretical result (Theorem 5.4) is almost verbatim a restatement of known results from Kidambi et al., 2021 on coverage-based sub-optimality bounds in for pessimistic offline RL. Fundamental quantities for the method to work, such as a definition for the uncertainty function u(s,a), are left under specified. Since u(s,a) drives the entire exploration method, leaving it abstract makes the method non-operational. In fact, obtaining a proper notion of uncertainty is well-studied and one of the defining challenges in reward-free RL (e.g. Amortila et al., 2024) and bonus-based exploration (Bellemare et al. 2016, Pathak et al. 2017, Pathak et al. 2019, Ash et al. 2022). And as mentioned before, though finding exploratory policies that cover the distribution of $\pi^\star$ is a good idea (since this would allow for subsequent offline RL), this is not a novel idea.

**Experimental limitations**

The experiments are limited to a toy gridworld and a supply-chain simulator, with very weak baselines (greedy and ε-greedy policies). No comparisons are provided to standard offline-to-online RL, reward-free methods, or bonus-based exploration as mentioned above. Without these, it is impossible to judge whether the proposed method adds practical value. Furthermore, the experiments do not seem to actually measure the deployment efficiency of their method. The results themselves seem modest over epsilon-greedy and not accompanied by ablations on the different choices of the uncertainty function.

Overall, the paper’s motivation is good, but the theoretical development is largely a repackaging of prior results, the new contributions are handwavy and left undefined, and the experiments lack strong baselines, ablations, and a quantitative measure of the claimed deployment efficiency.

Citations:
Bellemare et al. 2016: https://arxiv.org/pdf/1606.01868
Pathak et al. 2017: https://arxiv.org/pdf/1705.05363
Pathak et al. 2019: https://arxiv.org/pdf/1906.04161
Ash et al. 2022: https://arxiv.org/pdf/2110.11202
Jin et al. 2020: https://arxiv.org/pdf/2002.02794
Wang et al. 2020: https://arxiv.org/pdf/2006.11274
Chen et al. 2022: https://arxiv.org/pdf/2206.10770
Wagenmaker et al. 2022: https://arxiv.org/pdf/2201.11206
Amortila et al. 2024: https://arxiv.org/pdf/2403.06571

**Questions:**

- How should one compute the uncertainty function u(s,a) in practice outside of tabular domains?
- What differentiates Theorem 5.4 from prior results such as those in Kidambi et al. (2021)?
- How does your exploration method compare to prior reward-free methods, ensemble disagreement methods, curiosity-driven methods, or count-based exploration on the same tasks?

---

> ### Author Response · Authors · 2025-12-03
> **Author response**
>
> > Overall, the paper’s motivation is good, but the theoretical development is largely a repackaging of prior results,
> > main theoretical result is verbatim theoretical results from Kidambi et al 2021.
> > (e.g. Equation 1: \hat{p} is undefined, not specified which (s,a) is being considered),
>
> While Kidambi et al design a pessimistic MDP to solve for their algorithm and provide suboptimality bounds for a theoretical planner that is $\epsilon$-optimal on that MDP, we extend it to any offline RL algorithm trained on an offline dataset. We have been clear in the presented proofs about the lemmas taken from Kidambi et al. We would be happy to adjust the language to more clearly delineate our vs. Kidambi et al.’s results. The paper uses these results to find regions to explore **rather** than restrict themselves from venturing into these uncertain regions.
>
> $\hat{p}$ is defined in lines 153-154. The distribution is on state-actions so the $(s,a)$ considered here are from the state-action space of the MDP.
>
> > Fundamental quantities for the method to work, such as a definition for the uncertainty function u(s,a), are left under specified.
>
> We have defined u(s, a) in line 190. The goal of the paper is not to discuss the best uncertainty estimation and so we defined u(s, a) as the mathematically ideal estimator for uncertainty but the specific definitions will be dependent on the environment as can be seen in our supply-chain example.
>
> > No comparisons are provided to standard offline-to-online RL, reward-free methods, or bonus-based exploration as mentioned above.
>
> All these methods require network updates with online interactions. Specifically, offline-to-online methods update their policy as online samples are collected. The bonus based exploration methods also update their beliefs while collecting samples which affect the policy. We would like to re-iterate that in the real world setting that we are considering, it is not possible to perform ANY updates to any network that affects the verified action distribution of policy.
>
> > Furthermore, the experiments do not seem to actually measure the deployment efficiency of their method.
>
> Sorry for the confusion. The results in the gridworld and SupplyEnv domain demonstrate two distinct features of our exploration method. The former demonstrates that our method more quickly converges to the known pre-trained policy when performing rollouts near the datasets state-action distribution. The importance of these results is that they show our method only does exploration when there is dynamic uncertainty close to where the pre-trained policy is operating. While not directly demonstrating sample-efficiency through faster to the optimal policy (as shown in Fig. 2b), the exploration policy converges faster to the optimal policy (blue line in Fig. 2a).
>
> Secondly, the results in the SupplyEnv demonstrate a more traditional notion of deployment efficiency by training a policy that more quickly converges to the optimal policy; this is clearly shown by the superior performance of the policy trained after having collected new data using our method as opposed to the baselines (top row of Fig. 4).

---

### Meta-Review · Area_Chair_eKoF · 2026-01-10

**Summary:**

This paper proposes to use "deployment-efficient" reinforcement learning framework for safety-critical real-world settings, where agents must gather data using fixed, verifiable policies rather than updating continuously online. The method constructs a stationary exploration policy that maximizes an uncertainty metric while constrained to stay close to a pre-trained conservative policy, aiming to safely expand dataset coverage for subsequent offline training. While the problem setting is well-motivated, the reviewers unanimously feel the theoretical novelty is overstated (heavily borrowing from prior work without sufficient distinction), and the empirical baselines are too weak to justify the claims. The AC reviewed the rebuttal details and does not believe the paper is above the line of acceptance.

**Reviewer Concerns:**

*Addressed:* The authors clarified minor details regarding their experimental setup (e.g., the use of function approximation vs. tabular methods) and provided specific deployment metrics (trajectory counts) requested by Reviewer TFxf.

*Outstanding:*
- The authors justified excluding standard exploration baselines (like RND or Ensembles) by arguing that "online updates" are unsafe. However, this justification is unconvincing because it does not explain why they could not compare against stationary/frozen versions of these standard methods (e.g., an ensemble trained offline and fixed for deployment).

- The authors defended the similarity to Kidambi et al. (2021) by arguing that their work extends the theory to any offline algorithm. However, this relies heavily on Assumption 5.3 (that the offline learner finds the optimal policy on the pessimistic MDP). The rebuttal did not provide a strong argument for why this assumption is realistic (or tractable).

**Reviewer Scores:**

I would predict the reviewers might stay negative.

---

### Decision · Program_Chairs · 2026-01-26

Reject